# Management and outcomes of myelomeningocele-associated hydrocephalus in low-income and middle-income countries: a systematic review and meta-analysis protocol

Berjo Dongmo Takoutsing,[1,2] Alvaro Yanez Touzet [ID],[3] Jay J Park [ID],[4] Seong Hoon Lee [ID],[5] Emily R Bligh,[6,7] Abdullah Egiz,[8] Conor S Gillespie,[9] Anthony Figaji,[10,11] On behalf of the Neurology and Neurosurgery Interest Group

BDT and AYT contributed equally.

BDT and AYT are joint first authors.

For numbered affiliations see end of article.

**Correspondence to**
Mr Alvaro Yanez Touzet;
alvaro.yaneztouzet@student.manchester.ac.uk

## ABSTRACT

**Introduction** Hydrocephalus and myelomeningocele (MMC) place disproportionate burdens of disease on low-income and middle-income countries (LMICs). MMC-associated hydrocephalus and its sequelae result in a spectrum of severely devastating clinical manifestations, for which LMICs are disproportionately unprepared in terms of human, capital and technological resources. This study aims to review and compare the management and outcomes of infant MMC-associated hydrocephalus in LMICs and high-income countries.

**Methods and analysis** This systematic review and meta-analysis will follow the Preferred Reporting Items for Systematic Reviews and Meta-Analyses 2020 guidelines. The following databases will be searched without restrictions on language, publication date or country of origin: EMBASE, MEDLINE, The Cochrane Library, Global Index Medicus, African Journals Online and SciELO. All peer-reviewed studies of primary data reporting management and outcomes of infant MMC-associated hydrocephalus will be included. Where high-quality homogeneous studies exist, meta-analyses will be conducted to compare the management and outcomes of MMC-associated hydrocephalus across socioeconomic and geographical regions of the world. The primary outcome will be treatment failure of the first-line hydrocephalus treatment, which we defined operationally as the performance of a second intervention for the same reason as the first. Secondary outcomes include time to failure, rates of mortality and postoperative complications.

**Ethics and dissemination** Ethical approval was not applicable because this study does not involve human participants. Dissemination strategies will include publication in a peer-reviewed journal, oral and poster presentations at conferences and an interactive web application to facilitate interaction with the findings and promote the discussion and sharing of findings on social media.

**PROSPERO registration number** CRD42021285850.

## STRENGTHS AND LIMITATIONS OF THIS STUDY

⇒ This review focuses on multiple treatment modes of a well-defined disease population.
⇒ Six electronic databases that are commonly used across both high-income and low-income countries will be searched without restrictions on language, location or publication date.
⇒ The representativeness of the sample will rely on the quality of reporting of myelomeningocoele-associated hydrocephalus in the literature.
⇒ Only one operational definition of treatment failure—'the performance of a second intervention for the same reason as the first'—will be sought.
⇒ An interactive web application dashboard will be developed to facilitate the transparent interaction with our methods and findings and promote scientific discussion and scrutiny.

## INTRODUCTION

Hydrocephalus places a disproportionate burden of disease on low-income and middle-income countries (LMICs).[1 2] It affects approximately 1 in 1000 infants worldwide, but its incidence may exceed 200 000 cases per year in developing regions like sub-Saharan Africa.[3] Myelomeningocele (ie, meningo-myelocele or open spina bifida; MMC) is a common and severe spinal aetiology and constitutes a significant proportion of this population.[4–6] Its prevalence also varies by geography but approximates 113 cases per 100 000 births in LMICs,[7 8] reaching 77–610 and 700 cases per 100 000 births in South Africa and Nigeria, respectively. These disorders result in a spectrum of clinical manifestations, among which MMC-associated hydrocephalus is one of the most common and debilitating.[4 9 10]

In LMICs, the incidence of MMC-associated hydrocephalus is high and can affect as many as 75% of cases.[11 12] Times to diagnoses and treatment are often delayed in these settings and, if treatment is not promptly initiated, most patients do not survive beyond infancy.[4] Treatment may involve surgery, which is prone to significant morbidity and mortality, particularly in low-resource settings.[1 3] Intervening at the earliest stage with the best possible treatment remains, therefore, a crucial step in the management of MMC-associated hydrocephalus.

Ventriculoperitoneal (VP) shunts are the current standard of treatment in LMICs.[1] Increasingly, patients are being treated with endoscopic third ventriculostomy (ETV) or combinations of ETV and choroid plexus cauterisation (CPC), whose success rates and perceptions are variable.[1 3 4 9] Although ETV/CPCs were first developed and validated in LMICs as the primary management for MMC-associated hydrocephalus, there remains limited aggregated data regarding their outcomes in the very countries these procedures were developed in. As such, the practice and outcomes of these techniques in LMICs warrant systematic review and meta-analysis. Previous work has examined the management and outcomes of hydrocephalus in HICs and LMICs[13 14] although in unstratified medical subject headings age groups and aetiologies.

## STUDY AIMS AND OBJECTIVES

### Aim

This study aims to review and compare the management and outcomes of MMC-associated hydrocephalus in infants across countries and treatment modes.

### Objectives

To review and compare:
1. The first-line and second-line use of VP shunts, combinations of ETV and CPC, and conservative management.
2. The rates and times to treatment failure of VP shunts, combinations of ETV and CPC, and conservative management.
3. Measures of mortality and postoperative complications of VP shunts, combinations of ETV and CPC, and conservative management.

### Review questions

Primary question:
1. What are the rates of failure of VP shunts, ETV, combinations of ETV and CPC, and conservative treatments in infant MMC-associated hydrocephalus?

Secondary questions:
1. What are the most frequent first-line and second-line treatments in the management of infant MMC-associated hydrocephalus?
2. What is the time to failure of VP shunts, ETV, combinations of ETV and CPC, and conservative treatments in infant MMC-associated hydrocephalus?

3. What are the mortality and postoperative complication rates of VP shunts, ETV, combinations of ETV and CPC, and conservative treatments in infant MMC-associated hydrocephalus?
4. Are the management and outcomes of the MMC associated with the failure of the hydrocephalus treatment?

Subgroup analyses by country and treatment mode are planned (further details in Methods section). The primary outcome will be the rate of treatment failure of the first-line hydrocephalus treatment. Secondary outcomes include: time to failure, rates of mortality and postoperative complications. Operational definitions of outcomes are provided in the Methods section.

Due to its historical standing as the local standard of treatment, our principal hypothesis is that:

$H_1$: VP shunts have the lowest rates of treatment failure in the infant MMC-associated hydrocephalus population.

Secondary hypotheses include:

$H_2$: VP shunts are the most frequently used first-line treatment and have the lowest mortality and complication rates.

$H_3$: Resource constrained environments are associated with late/worse MMC presentation among infants.

$H_4$: Late/worse MMC presentation is associated with high MMC treatment failure, mortality and complication rates.

$H_5$: MMC complication rates are associated with hydrocephalus treatment failure, mortality and complication rates.

## METHODS

This systematic review will be conducted following the guidelines outlined by the Preferred Reporting Items for Systematic Reviews and Meta-Analyses (PRISMA) 2020 statement.[15]

### Search

A search strategy was developed to identify research articles on MMC-associated hydrocephalus. This was adapted from McCarthy *et al*, and consisted of synonyms of 'hydrocephalus' and 'myelomeningocele' (online supplemental table 1). The search was run on the following electronic databases, from inception to 5 October 2021: EMBASE, MEDLINE, The Cochrane Library, Global Index Medicus, African Journals Online and ScieLO. No restrictions on language, location or publication date were placed, and unpublished studies will not be sought. Searches in all databases will be rerun prior to final analysis.

### Study selection

#### Types of studies

We will include articles published in peer-reviewed journals with any of the following designs: original research, trials, cross-sectional and cohort studies, multiple case reports (ie, >1 case) and case series. Opinion pieces, comments, letters, guidelines, editorials, single-case reports, reviews, meta-analyses and qualitative studies will

be excluded, as well as articles published in non-peer-reviewed journals.

### Country income level

We will include studies whose data were collected in low-income, lower-middle-income, upper-middle-income and high-income economies, according to the World Bank Country and Lending Groups.[16] Studies from high-income economies will be used as comparators. Studies whose data cannot be traced to one particular country will be excluded.

### Types of participants

All studies of infants aged 2 years or younger with a clinical or imaging diagnosis of MMC-associated hydrocephalus will be included. Non-infants (ie, >2 years) will be excluded, as will infants with diagnoses of lipomyelomeningocoele, spina bifida occulta or unspecified spina bifida, regardless of association with hydrocephalus.

### Types of interventions

Studies of infants who have undergone either surgical or conservative treatment for hydrocephalus will be included. Surgical treatment includes VP shunting, ETV, combined ETV and CPC, combined VP shunting and ETV, and combined VP shunting, ETV and CPC. We defined conservative treatment as non-surgical medical or pharmacological treatment. Studies of infants who underwent treatment for MMC, but not hydrocephalus, will be excluded, as will patients who received neither surgical nor conservative interventions.

### Types of outcome measures

We will include studies of primary data reporting measures of treatment failure. In this review, treatment failure will be operationally defined as the performance of a second intervention for the same reason as the first. Studies reporting measures of mortality, morbidity, postoperative complications and follow-up duration will also be included for secondary analysis. Postoperative complications will be defined as the unfavourable result of an intervention that did not result in treatment failure, and mortality as either 'intraoperative', if death occurred during a procedure, or 'perioperative', if it occurred within 30 days of surgery. For conservative treatments, mortality will encompass deaths that occurred within 30 days of administration.

### Outcomes

The primary outcome of our study will be the rate of treatment failure for the first-line hydrocephalus treatment. Secondary outcomes will include time to failure, rates of mortality and postoperative complications.

### Study selection

Search results will be uploaded to Rayyan (https://www.rayyan.ai) to facilitate deduplication and independent, blinded screening.[17] First, titles and abstracts will be screened by two independent reviewers against the inclusion criteria, and the eligibility of the selected abstracts will be determined by reading the full texts. Unless otherwise stated, conflicts will be resolved through discussion and, where consensus cannot be achieved, through arbitration by the senior author (AF).

### Data extraction

Two reviewers will independently extract and check the data extracted from the included studies using a standardised extraction proforma in Google Sheets (online supplemental table 2). Any disputes will be settled by a third reviewer (BDT, AYT or AF). Data items will include information on study and sample characteristics; vertebral level of MMC and aetiology of hydrocephalus; clinical presentation and method of diagnosis; treatment mode, timing and follow-up; and primary and secondary outcomes. If data are insufficiently provided in the full text, we will contact the corresponding author to request for the missing information and wait up to two months for a response. Following correspondence, all available data will be reported and studies with missing data will not be eligible for inclusion in the meta-analysis. All data will be recorded and stored in a spreadsheet.

### Risk of bias assessment

Included studies will be assessed for risk of bias by two independent reviewers of the extraction team. Randomised studies will be assessed using Version 2 of the Cochrane risk-of-bias tool for randomised trials (RoB 2).[18] Non-randomised studies will be assessed using the ROBINS I tool.[19] Risk of bias assessment will be used during the analysis and results presented in terms of the primary and secondary objectives of this study.

### Data analysis

Following extraction, data will be transferred to Python for analysis. Screening results will be presented using a PRISMA flow diagram. Study and sample characteristics will then be summarised using descriptive statistics; these may include: vertebral level of MMC; aetiology of hydrocephalus; method of diagnosis; treatment mode; indication for surgery (if surgical management); timing and follow-up; and primary and secondary outcomes.

The use, failure and mortality measures of VP shunts, combinations of ETV and CPC, and conservative management will then be meta-analysed. Although meta-analyses are planned, these will only become apparent after data extraction is complete. Publication bias will be assessed through the use of funnel plots and Egger's test.[20] If asymmetry is found in the funnel plot, the Trim-and-Fill method will be used to account for any potential publication bias.[21] The overall estimates and 95% CIs will be obtained using random-effects models as per established methods.[22] Cochran's Q test (p<0.10) and the $I^2$ statistic will be used to assess heterogeneity among studies.[23] Unless otherwise stated, the statistical significance will be set at p<0.05, and comparisons to high-income economies will be drawn.

Mortality, failure and complication rates will however vary with hydrocephalus aetiology, age, mode of treatment, characteristics of MMC, socioeconomic region and crucially, duration of follow-up. Therefore, subgroup analyses will be sought, as appropriate, and particular care will be taken when interpreting failure between shunt placement and endoscopic treatments, owing to contrasting patterns of failure over the shorter and longer terms. A combination of dichotomous, proportion, continuous, O–E and variance meta-analyses are planned for this purpose, as appropriate, depending on the data, and may include a potential subgroup analysis of the primary outcome into short-term rates and long-term rates. Recognising that case reports may present unusual and/or protracted outcomes, and randomised trials may assess active interventions improving outcomes, analyses by study design are also planned.

### Strength of body of evidence

The confidence in cumulative evidence included in this review will be assessed using the GRADE approach. Two independent reviewers will use the GRADEpro Guideline Development Tool (GRADEpro) to assess the quality of outcomes of this study. A third reviewer (BDT, AYT or JJP) will settle disagreements.

### Ethics and dissemination

Ethics approval for this study was not applicable because this study did not involve human participants. Dissemination strategies will include publication in a peer-reviewed journal, oral and poster presentations at conferences, and an interactive web application to facilitate interaction with the findings and promote the discussion and sharing of findings on social media.

### Patient and public involvement

None

**Author affiliations**
[1]Faculty of Health Sciences, University of Buea, Buea, Cameroon
[2]Research Department, Association of Future African Neurosurgeons, Yaounde, Cameroon
[3]School of Medical Sciences, Faculty of Biology, Medicine and Health, University of Manchester, Manchester, United Kingdom, University of Manchester, Manchester, UK
[4]Edinburgh Medical School, The University of Edinburgh, Edinburgh, UK
[5]Academic Critical Care & Neurosurgery, Aberdeen Royal Infirmary, Aberdeen, UK
[6]Surgical Specialties, The Queen Elizabeth University Hospital, Glasgow, UK
[7]Academic Foundation Programme, The University of Glasgow, Glasgow, UK
[8]School of Medicine, University of Central Lancashire, Preston, UK
[9]Institute of Systems, Molecular and Integrative Biology, University of Liverpool, Liverpool, UK
[10]Division of Paediatric Neuroscience (Neurosurgery), School of Child and Adolescent Health, University of Cape Town, Cape Town, South Africa
[11]Red Cross Children's Hospital, Cape Town, South Africa

**Acknowledgements** We thank Moniba Korch, Brian Ou Yong Ming, and Safia Farsana Shabeer for their contributions during the development and piloting of the extraction proforma.

**Contributors** BDT, AYT and AF were responsible for conceiving the article. AF is the guarantor. BDT, AYT, JJP and CSG wrote the manuscript. AE, AF, BDT, AYT, CSG, ERB, JJP and SHL provided a critical appraisal of the manuscript. All authors critically

revised and approved the final manuscript. BDT and AYT contributed equally and are joint first authors of the manuscript.

**Funding** The authors have not declared a specific grant for this research from any funding agency in the public, commercial or not-for-profit sectors.

**Competing interests** None declared.

**Patient and public involvement** Patients and/or the public were not involved in the design, or conduct, or reporting, or dissemination plans of this research.

**Patient consent for publication** Not applicable.

**Provenance and peer review** Not commissioned; externally peer reviewed.

**ORCID iDs**
Alvaro Yanez Touzet http://orcid.org/0000-0001-9309-1885
Jay J Park http://orcid.org/0000-0001-8762-6986
Seong Hoon Lee http://orcid.org/0000-0002-5210-4923

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
