## [Reviewer comments · BMJ Open]

ARTICLE DETAILS

TITLE (PROVISIONAL)	Management and outcomes of myelomeningocele-associated hydrocephalus in low- and middle-income countries: a systematic review and meta-analysis protocol
AUTHORS	Takoutsing, Berjo Dongmo; Yanez Touzet, Alvaro; Park, Jay; Lee, Seong Hoon; Bligh, Emily; Egiz, Abdullah; Gillespie, Conor; Figaji, Anthony; Interest Group, Neurology and Neurosurgery

VERSION 1 – REVIEW

REVIEWER	Surti, Ambreen Bahria University Medical and Dental College, Anatomy, Medical Education
REVIEW RETURNED	17-Oct-2022

GENERAL COMMENTS	A well conducted and written meta-analysis which is much needed in this field
---

REVIEWER	Warf, Benjamin C. Boston Children's Hospital, Neurosurgery
REVIEW RETURNED	30-Oct-2022

GENERAL COMMENTS	The authors propose a systematic review and meta-analysis of the literature in regard to management and outcomes of hydrocephalus associated with myelomeningocele in low- and middle-income countries. Their stated primary aim is to review and compare the modes of treatment and their respective outcomes for this condition among different countries. Results from low-middle income countries (LMIC) will be compared to those reported from high-income countries (HIC). This intended review is commendable and potentially valuable. I would like to make suggestions along three lines that would be important to strengthen this study: 1) the authors' perspective of endoscopic third ventriculostomy (ETV) with or without choroid plexus cauterization (CPC) in relation to LMIC; 2) a relevant interpretation of treatment failure; and, 3) the concept of "conservative" management. First, the Introduction section indicates the authors' intention to determine to what degree ETV or ETV/CPC, which they perceive – as a given – to be more prevalent in HIC, are being adopted by LMIC. In reality, ETV/CPC as the primary treatment of myelomeningocele-associated hydrocephalus was actually developed in the country of Uganda over two decades ago. The procedure has subsequently been adopted by many centers in North America; however, both prior to and during the same time
--

period it was being adopted at specific sites elsewhere in the world in countries that included Tanzania, Malawi, Zambia, Angola, Nigeria, Mali, Vietnam, Bangladesh, Indonesia, the Philippines, and Iran. The ETV/CPC procedure is actually an unusual and counterintuitive case in which a new treatment was developed and validated in a low-income country and subsequently exported to a high-income region of the world (East to West and South to North). In light of the foregoing, a reframing of the context and intent of the proposed study is needed.

Second, in the Study Aims and Objectives section, the authors shift their emphasis more toward a direct comparison of shunt placement to endoscopic treatment in regard to rates of failure. Specifically, treatment failure is noted to be the “primary question” and the “primary outcome” of the study. Their stated principal hypothesis is that VP shunts have the lowest rates of treatment failure for this etiology of hydrocephalus and, secondarily, that shunt placement has the lowest rates of mortality and complications. Although the authors do intend to look at time to treatment failure as a secondary outcome, they need to be very careful and clear about the intended minimum time of follow up required of the studies included in their analysis. It is crucial to recognize the contrasting patterns of failure between shunt placement and endoscopic treatment. Unlike shunts, the great majority of all ETV and ETV/CPC failures will have occurred within 6 months of the procedure, with subsequent failures becoming increasingly rare. Shunts, on the other hand, will continue to fail over time, with 60-80% having failed at least once in less than 10 years, and with multiple failures over time being the norm. Furthermore, when treating infants, the clinical consequences of early treatment failure are not as precipitous or life-threatening as in the case of children later on when the condition of shunt-dependence is potentially fatal. This is a crucially important concept in low-resource settings, where quick access to emergency neurosurgical care for shunt failure typically does not exist. Under these circumstances, it is the “long game” that takes precedence when considering an optimum public health strategy for infant hydrocephalus. Any meaningful comparison of treatment failure between these modalities must take these considerations into account. Thus, the authors will need to filter out any short-term comparisons of failure rates, which may prove difficult due to the challenges of longer-term patient follow up in LMIC.

Finally, the concept of “conservative management” in the proposed review needs to be better defined. On the one hand, the authors state that conservative management outcomes will be assessed, while stating on the other hand that cases of myelomeningocele without hydrocephalus treatment will be excluded. This begs the question of how “conservative management” differs from cases when infants were not treated for hydrocephalus. Evaluating this particular category of patients is further complicated by the fact that children with myelomeningocele may have insidiously progressive hydrocephalus, may develop a “compensated” hydrocephalus, or have hydrocephalus that is clinically manifested more by the effects of syringobulbia or hydromyelia that might not have been properly recognized as resulting from untreated hydrocephalus.

	The authors are to be commended for proposing this review and analysis. The three areas of concern outlined above will need to be adequately addressed.
--	---

VERSION 1 – AUTHOR RESPONSE

Reviewer: 1

Dr. Ambreen Surti, Bahria University Medical and Dental College

Comments to the Author:

A well conducted and written meta-analysis which is much needed in this field

Our response: Thank you very much for your appreciation.

Reviewer: 2

Dr. Benjamin C. Warf, Boston Children's Hospital

Comments to the Author:

The authors propose a systematic review and meta-analysis of the literature in regard to management and outcomes of hydrocephalus associated with myelomeningocele in low- and middle-income countries. Their stated primary aim is to review and compare the modes of treatment and their respective outcomes for this condition among different countries. Results from low-middle income countries (LMIC) will be compared to those reported from high-income countries (HIC).

This intended review is commendable and potentially valuable. I would like to make suggestions along three lines that would be important to strengthen this study: 1) the authors' perspective of endoscopic third ventriculostomy (ETV) with or without choroid plexus cauterization (CPC) in relation to LMIC; 2) a relevant interpretation of treatment failure; and, 3) the concept of "conservative" management.

Our response: Thank you very much for your commendation. We very much welcome any and all suggestions.

First, the Introduction section indicates the authors' intention to determine to what degree ETV or ETV/CPC, which they perceive – as a given – to be more prevalent in HIC, are being adopted by LMIC. In reality, ETV/CPC as the primary treatment of myelomeningocele-associated hydrocephalus was actually developed in the country of Uganda over two decades ago. The procedure has subsequently been adopted by many centers in North America; however, both prior to and during the same time period it was being adopted at specific sites elsewhere in the world in countries that included Tanzania, Malawi, Zambia, Angola, Nigeria, Mali, Vietnam, Bangladesh, Indonesia, the Philippines, and Iran. The ETV/CPC procedure is actually an unusual and counterintuitive case in which a new treatment was developed and validated in a low-income country and subsequently exported to a high-income region of the world (East to West and South to North). In light of the foregoing, a reframing of the context and intent of the proposed study is needed

Our response: Thank you for raising this valuable point. A reframing of the context and intent has been made in the Introduction. Indeed, myelomeningocele-associated hydrocephalus is a unique case whereby a surgical intervention has been initially developed and validated in a developing country—and, at the same time, where systematic review and quantification of its outcomes, within those same countries, is still needed. We now acknowledge both the early work of LMIC based authors on interventions such as ETV/CPC, and that LMICs have the highest caseload of MMC worldwide, throughout the revised manuscript.

Second, in the Study Aims and Objectives section, the authors shift their emphasis more toward a direct comparison of shunt placement to endoscopic treatment in regard to rates of failure. Specifically, treatment failure is noted to be the "primary question" and the "primary outcome" of the

study. Their stated principal hypothesis is that VP shunts have the lowest rates of treatment failure for this etiology of hydrocephalus and, secondarily, that shunt placement has the lowest rates of mortality and complications. Although the authors do intend to look at time to treatment failure as a secondary outcome, they need to be very careful and clear about the intended minimum time of follow up required of the studies included in their analysis. It is crucial to recognize the contrasting patterns of failure between shunt placement and endoscopic treatment. Unlike shunts, the great majority of all ETV and ETV/CPC failures will have occurred within 6 months of the procedure, with subsequent failures becoming increasingly rare. Shunts, on the other hand, will continue to fail over time, with 60-80% having failed at least once in less than 10 years, and with multiple failures over time being the norm.

Our response: Thank you for this very insightful comment. We wholeheartedly agree on the necessity to be careful and clear when interpreting failure between shunt placement and endoscopic treatment, owing to contrasting patterns of failure. To this end, we have made an explicit note of this potential confounder in lines 300-303, to guarantee their visibility and incorporation into data analysis and interpretation within this protocol. We will absolutely consider this in the analysis.

Furthermore, when treating infants, the clinical consequences of early treatment failure are not as precipitous or life-threatening as in the case of children later on when the condition of shunt-dependence is potentially fatal. This is a crucially important concept in low-resource settings, where quick access to emergency neurosurgical care for shunt failure typically does not exist. Under these circumstances, it is the “long game” that takes precedence when considering an optimum public health strategy for infant hydrocephalus. Any meaningful comparison of treatment failure between these modalities must take these considerations into account. Thus, the authors will need to filter out any short-term comparisons of failure rates, which may prove difficult due to the challenges of longer-term patient follow up in LMIC.

Our response: Once again, thank you for another very insightful comment. As the reviewer points out, failure rates must be contextualised in time, and one meaningful way of doing so is to filter out short-term rates to focus on the long game, whose advantage is to inform optimum public health strategies for infant hydrocephalus. We agree that the limited long-term duration of follow-ups across LMICs may limit our ability to focus on the long-term rates exclusively, and thus, propose in lines 305-306 to conduct a subgroup analysis of both short and long-term rates, contingent on the quality of the data from this body of literature.

Finally, the concept of “conservative management” in the proposed review needs to be better defined. On the one hand, the authors state that conservative management outcomes will be assessed, while stating on the other hand that cases of myelomeningocele without hydrocephalus treatment will be excluded. This begs the question of how “conservative management” differs from cases when infants were not treated for hydrocephalus.

Our response: Thank you for raising this point. We have made our definition of conservative management more objective in lines 218-219 (“non-surgical medical or pharmacological treatment”) and clarified in lines 220-221 that it is patients who received neither surgical nor conservative management who will be excluded.

Evaluating this particular category of patients is further complicated by the fact that children with myelomeningocele may have insidiously progressive hydrocephalus, may develop a “compensated” hydrocephalus, or have hydrocephalus that is clinically manifested more by the effects of syringobulbia or hydromyelia that might not have been properly recognized as resulting from untreated hydrocephalus.

Our response: Thank you for this comment. It is certainly a limitation of this study design that the representativeness of the sample will depend on the quality of reporting of infant myelomeningocele-associated hydrocephalus in the literature. We have thus added a bullet point about this under the

section “Strengths and limitations of this study”, to acknowledge the limitation in a forthcoming manner.

The authors are to be commended for proposing this review and analysis. The three areas of concern outlined above will need to be adequately addressed.

Our response: Thank you once again for your commendation. We have incorporated major changes to the manuscript as a result of these observations and hope to have addressed these three areas adequately.

VERSION 2 – REVIEW

REVIEWER	Warf, Benjamin C. Boston Children's Hospital, Neurosurgery
REVIEW RETURNED	19-Dec-2022

GENERAL COMMENTS	This intended review is commendable and potentially valuable. In my review of the original submission, I made suggestions along three lines that would be important to strengthen this study: 1) the authors' perspective of endoscopic third ventriculostomy (ETV) with or without choroid plexus cauterization (CPC) in relation to LMIC; 2) a relevant interpretation of treatment failure; and, 3) the concept of “conservative” management. The authors have responded to each. First, I had pointed out that the Introduction section indicates the authors' intention to determine to what degree ETV or ETV/CPC, which they perceive – as a given – to be more prevalent in HIC, are being adopted by LMIC. In reality, ETV/CPC as the primary treatment of myelomeningocele-associated hydrocephalus was actually developed in the country of Uganda over two decades ago. The procedure has subsequently been adopted by many centers in North America; however, both prior to and during the same time period it was being adopted at specific sites elsewhere in the world in countries that included Tanzania, Malawi, Zambia, Angola, Nigeria, Mali, Vietnam, Bangladesh, Indonesia, the Philippines, and Iran. The ETV/CPC procedure is actually an unusual and counterintuitive case in which a new treatment was developed and validated in a low-income country and subsequently exported to a high-income region of the world (East to West and South to North). In light of the foregoing, a reframing of the context and intent of the proposed study is needed In the authors' response, they stated that "systematic review and quantification of its outcomes, within those same countries (i. e. the LMIC countries that have adopted the ETV/CPC procedure), is still needed". This is certainly true in regard many individual countries (as will be the case for adequate studies of any modality of treatment), but this deficit will not be accomplished by a meta-analysis of the existing published literature. However, this meta-analysis should be able to identify those studies with reliable data and put them into context, allowing for a higher-level overall view. Areas requiring further investigation will, hopefully, be highlighted, as well as those demonstrating concordance amongst published investigations.
---

Second, in the Study Aims and Objectives section, I had noted that the authors shift their emphasis more toward a direct comparison of shunt placement to endoscopic treatment in regard to rates of failure. Specifically, treatment failure is noted to be the “primary question” and the “primary outcome” of the study. Their stated principal hypothesis is that VP shunts have the lowest rates of treatment failure for this etiology of hydrocephalus and, secondarily, that shunt placement has the lowest rates of mortality and complications. Although the authors do intend to look at time to treatment failure as a secondary outcome, I stressed that they need to be very careful and clear about the intended minimum time of follow up required of the studies included in their analysis. It is crucial to recognize the contrasting patterns of failure between shunt placement and endoscopic treatment. Unlike shunts, the great majority of all ETV and ETV/CPC failures will have occurred within 6 months of the procedure, with subsequent failures becoming increasingly rare. Shunts, on the other hand, will continue to fail over time, with 60-80% having failed at least once in less than 10 years, and with multiple failures over time being the norm. Furthermore, when treating infants, the clinical consequences of early treatment failure are not as precipitous or life-threatening as in the case of children later on when the condition of shunt-dependence is potentially fatal. This is a crucially important concept in low-resource settings, where quick access to emergency neurosurgical care for shunt failure typically does not exist. Under these circumstances, it is the “long game” that takes precedence when considering an optimum public health strategy for infant hydrocephalus. Any meaningful comparison of treatment failure between these modalities must take these considerations into account. Thus, the authors will need to filter out any short-term comparisons of failure rates, which may prove difficult due to the challenges of longer-term patient follow up in LMIC.

In their response, it seems that the authors have agreed to be intentional in recognizing the extreme limitations of drawing any clinically meaningful conclusions from short-term results, given the known differences in failure patterns between shunt placement and endoscopic treatment. Indeed, the authors committed to conduct a subgroup analysis of both short and long-term outcomes, "contingent on the quality of the data from this body of literature".

Finally, I had noted that the concept of “conservative management” in the proposed review needs to be better defined. On the one hand, the authors state that conservative management outcomes will be assessed, while stating on the other hand that cases of myelomeningocele without hydrocephalus treatment will be excluded. This begs the question of how “conservative management” differs from cases when infants were not treated for hydrocephalus. I also noted that evaluating this particular category of patients is further complicated by the fact that children with myelomeningocele may have insidiously progressive hydrocephalus, may develop a “compensated” hydrocephalus, or may have hydrocephalus that is clinically manifested more by the effects of syringobulbia or hydromyelia that might not have been properly recognized as resulting from untreated hydrocephalus.

The authors responded to these comments by defining conservative management more objectively as “non-surgical

	medical or pharmacological treatment". They further clarified that patients who received neither surgical nor "conservative" (i.e. medical) management will be excluded from their outcomes analysis. They also have included a statement in their "Strengths and limitations of the study" regarding the potential diagnostic nuances in this population, that might limit their conclusions. The authors are to be commended for proposing this review and analysis, and the three areas of concern outlined above have been reasonably addressed.
--	--